# The effect of temperature on the boundary conditions of West Nile virus circulation in Europe

**Eduardo de Freitas Costa[1], Kiki Streng[2], Mariana Avelino de Souza Santos[1], Michel Jacques Counotte**[1]*

**1** Wageningen Bioveterinary Research, Wageningen University and Research, Lelystad, the Netherlands, **2** Quantitative Veterinary Epidemiology, Wageningen University and Research, Wageningen, the Netherlands

* michel.counotte@wur.nl

**Data Availability Statement:** All data is available on the git of the project: https://git.wur.nl/wbvr_epi/WNVEurope.

## Abstract

West Nile virus (WNV) is a vector-borne flavivirus that causes an increasing number of human and equine West Nile fever cases in Europe. While the virus has been present in the Mediterranean basin and the Balkans since the 1960s, recent years have witnessed its northward expansion, with the first human cases reported in Germany in 2018 and the Netherlands in 2020. WNV transmission and amplification within mosquitoes are temperature-dependent. This study applies a mathematical modelling approach to assess the conditions under which WNV circulation occurs based on the proportion of mosquito bites on WNV-competent birds (dilution), vector-host ratios, mosquito season length and the observed daily temperature data. We modelled five distinct European regions where previous WNV circulation has been observed within the Netherlands, Germany, Spain, Italy, and Greece. We observed that the number of days in which the basic reproduction number ($R_0$) is above one, increased over the last 40 years in all five regions. In the Netherlands, the number of days in which the $R_0$ is above one, is 70% lower than in Spain. The temperature in Greece, Spain and Italy allowed for circulation under low vector-host ratios, and at a high dilution. On the other hand in the Netherlands and Germany, given the observed daily temperature, the thresholds for circulation requires a lower dilution and higher vector-host ratios. For the Netherlands, a short window of introductions between late May and mid-June would result in detectable outbreaks. Our findings revealed that the temperate maritime climate of the Netherlands allows WNV circulation primarily during warmer summers, and only under high vector-host ratios. This research contributes valuable insights into the dynamic relationship between temperature, vector properties, and WNV transmission, offering guidance for proactive strategies in addressing this emerging health threat in Europe.

## Author summary

West Nile virus (WNV) is a virus that is transmitted by mosquitoes, and that leads to West Nile fever cases in humans and horses. Although the virus has been in certain parts

**Funding:** This project was funded by the Dutch Ministry of Agriculture, Nature and Food Quality (RA-vector overdraagbare ziekten, WOT-KB-37-003-053 to MC, EC and MA. The funders had no role in study design, data collection and analysis, decision to publish, or preparation of the manuscript.

**Competing interests:** The authors have declared that no competing interests exist.

of Europe since the 1960s, it has recently spread northward, with the first human cases reported in Germany in 2018 and the Netherlands in 2020. Here, we use mathematical modelling to understand the conditions necessary for WNV transmission. We consider the proportion of mosquito bites on birds that can carry the virus, the number of hosts and vectors, the length of mosquito season, and temperature data in specific European regions where WNV has been observed previously (Netherlands, Germany, Spain, Italy, and Greece). We found that the number of days suitable for WNV transmission (determined by the basic reproduction number, R0) has increased over the last 40 years in all these regions. Furthermore, we found differences among these regions. In the Netherlands, for instance, the conditions for virus circulation requires higher vector-host ratios, while in warmer regions like Greece, Spain, and Italy, circulation occurs under lower ratios. Understanding the connection between temperature, mosquito traits, and WNV transmission is crucial for implementing surveillance and preventive measures.

## Introduction

West Nile virus (WNV) is a flavivirus that has an amplifying cycle between mosquitoes and birds. Birds are the natural reservoirs for WNV and susceptible species can develop high levels of viraemia, allowing the virus to be transmitted to mosquitoes that feed on them [1]. WNV can be transmitted to horses and humans through the bite of infected mosquitoes, however, these animals are unable to propagate infection and are considered dead-end hosts. Infection in horses and humans can result in flu-like illness, that can further complicate into neurological disease. WNV virus was first identified in the West Nile region of Uganda in 1937 and has since spread to other parts of the world, where now the virus is endemic in Africa, Europe, the Middle East, North America and West Asia [2,3].

In Europe, the occurrence of the virus exhibits distinct spatial patterns, where some regions are heavily affected and others less so. In Southern Europe, WNV has been circulating for several decades and human cases are reported during the summer and early autumn. For example, in Spain the Doñana National Park acts as a hotspot for WNV circulation [4]. In Italy, the Po Valley in the northern part of the country is an important region where WNV is endemic [5,6]. Another example is the Aksiou Delta, a rice-growing region in western Thessaloniki in Greece [7]. These areas contribute to 58% of the total reported human WNV cases within Europe, between 2010–2021 [8]. In Europe, ornithophilic species of the complex *Culex (Cx.) pipiens s.l.* are the most common vector species for WNV [9].

Since 2018, WNV outbreaks have occurred across larger geographical areas in Europe, for the first time also affecting northern countries including Germany (2018) and the Netherlands (2020) [10]. During the summer of 2018, the first introduction of the virus in Germany was detected [11], which managed to over-winter. In the next year, 2019, 76 cases in birds, 36 in horses, and five confirmed mosquito-borne, autochthonous human cases were reported [11]. It was believed that the exceptionally warm summer of 2018 facilitated the emergence and establishment of the disease [12]. In August 2020, a common whitethroat (*Curruca communis*) was caught in the municipality of Utrecht, the Netherlands and tested positive for WNV lineage 2 [13]. Similarly, two of 44 pools of *Cx. pipiens* mosquitoes tested positive for WNV RNA, confirming local circulation of the virus. In October 2020, for the first time, an autochthonous human case of WNV was diagnosed [14]. Active case finding identified another five cases in Utrecht, and one near Arnhem. In the subsequent year no WNV circulation was detected, despite active surveillance efforts. In the late summer of 2022, a grey heron (*Ardea cinerea*)

tested positive for WNV [15]. It has been demonstrated that *Culex* mosquitoes in Germany and the Netherlands are competent WNV vectors [16,17].

The epidemiology of WNV is partly driven by temperature. Especially the exceptionally warm summer of 2018 has been thought to be the driving force behind the equally exceptionally large European outbreak [12,18]. It has been hypothesised that climate change will result in a greater risk of introduction, transmission and overwintering of WNV in Europe [3]. This can be explained by the fact that mosquitoes, the virus and their interaction are partly governed by temperature [19–26] and average temperatures are going to increase. Important life-cycle parameters such as lifespan [19–21], biting rate [19,22–24], reproduction rate and mosquito activity are temperature dependent [27]. Similarly, viral establishment, replication and incubation period within mosquitoes are driven by temperature [25,26]. By combining these parameters, a thermal optimum of WNV transmission through *Cx. pipiens* is thought to be around 24.5 (23.6–25.5) degrees Celsius, and transmission can occur between 16.8 and 34.9 degrees Celsius [27]. Climate change can thus expand the areas where temperature is suitable for sustained WNV circulation.

Early detection is key to implement preventive measures, such as vector control measures and laboratory screening of blood and organ donations [2,28]. Surveillance is crucial to allow early detection of the virus. Surveillance efforts involve monitoring mosquito populations, animal hosts and humans [2,5]. Integrated One Health strategies and (risk-based) surveillance leverage the possibility of combining surveillance in multiple species [5]. In the EU, this integrated surveillance generally consists of sampling of wild birds and mosquitoes, combined with passive surveillance (notification of cases) in horses and humans. Some countries implement additional activities such as active sampling of horses, sentinel chickens or sampling of other (wildlife) species such as dogs, deer or wild boar [29]. According to EU law, cases in equids and humans are notifiable. Effective surveillance and early detection can further aid in identifying areas that are at higher risk for WNV outbreaks, allowing for targeted intervention and resource allocation [30].

Mathematical modelling can help estimate under which conditions WNV transmission can take place and thus inform surveillance strategies [31]. Several models have implemented temperature-dependent parameters into disease transmission models [12,32–34]. Mathematical models can be valuable tools for understanding disease dynamics and be used as tool for risk management for planning and optimizing surveillance efforts for WNV. These models may assist in predicting the spatial and temporal dynamics of WNV transmission, identifying high-risk areas, and assessing the effectiveness of different control strategies [32,35].

In this paper, we propose a compartmental model that allows us to explore under which conditions WNV transmission between mosquitoes and birds takes place. We used a mechanistic mathematical model to assess the combined effect of temperature, proportion of competent vectors and host, and the number and time of WNV introductions on the boundary conditions for WNV circulation.

## Methods

### Concepts and assumptions

We consider five locations in Europe, based on their previous circulation of WNV. In Spain, we consider the Doñana Natural Park [4,36], in Italy the Po Valley [6], in Greece the rice-growing region of western Thessaloniki [7], in Germany the area of Berlin [11], and in the Netherlands the area around Utrecht [14]. The Nomenclature of Territorial Units for Statistics (NUTS) two regions in which these places are located, accounted for 58% of the reported human West Nile fever cases in Europe over the last decade (S1 Text).

In the proposed compartmental model, we distinguish three concepts that will vary within and between different model settings:

1. The abundance of mosquitoes relative to the number of hosts (vector-host ratio), is defined here as the vector-host ratio as the number of mosquitoes per bird.

2. The proportion of mosquitoes' bites on WNV-competent birds (dilution) as the proportion of susceptible and competent birds that are able to replicate the virus to infect mosquitoes. Dilution thus corrects for the number of mosquito bites that are either on non-susceptible birds, or even other (non-amplifying) hosts.

3. The duration of the mosquito season, as defined with a day of onset and a duration in number of days during which mosquitoes are active.

In the model, we assumed that 1) birds' mortality is not affected by the disease status, 2) mosquitoes and birds' populations are constant over time, 3) all hatched mosquitoes (births) are susceptible, and 4) the vector-host ratio is constant over the season; 5) due to the temperature dependence of the metabolism of mosquitoes and thus viral replication, the parameters within the mosquito compartment are considered to be temperature-dependent variables.

## Model description

We extended the model described by Wonham et al. (2004) [31] by adding temperature dependence to the parameters. In the model, mosquitoes can exist in three different states: Active biting mosquitoes in which transmission dynamics occur: $S_M$, $E_M$, and $I_M$ (susceptible, latently infected, and infectious adult mosquitoes), and birds can exist in either a susceptible ($S_B$), infectious ($I_B$), or recovered ($R_B$) state. Individuals move between compartments according to specific rates (Fig 1).

The individuals change between compartments following different rates: the biting rate ($a$) and dilution ($d$) are multiplied either with the per contact transmission probability mosquito-to-bird ($b$), moving individuals from ($S_B$) to ($I_B$), or per contact transmission probability bird-to-mosquito ($c$), moving individuals from ($S_M$) to ($E_M$). Birds get recovered from infection at the bird recovery rate ($g$). Mosquitos move from ($E_M$) to ($I_M$) according the incubation rate ($k$) or median duration of latent period ($1/k$, for mosquitoes). Birds are replaced at a rate ($\mu_V$), where $1/\mu_V$ is the average lifespan of the birds, and mosquitoes are replaced at a rate $\mu_A$, where

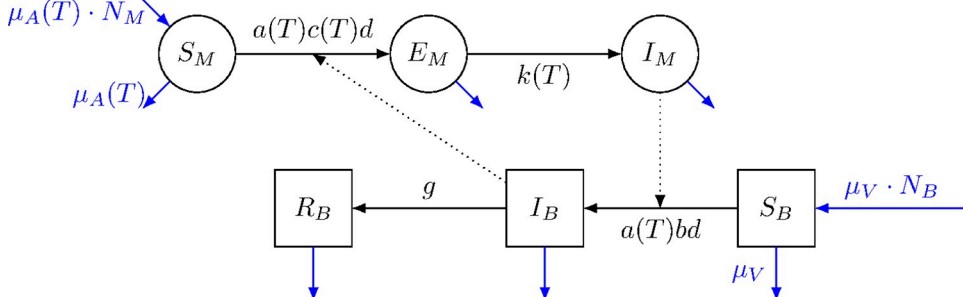

**Fig 1. Schematic representation of the compartmental transmission model.** Mosquitoes (circles) can exist as susceptible ($S_M$), latently infected ($E_M$) and infectious ($I_M$); Birds (squares) can exist as susceptible ($S_B$), infectious ($I_B$), or recovered ($R_B$), moving between compartments according to different rates. Total number of mosquitoes and the total number of birds are provided by $N_M$ and $N_B$. The biting rate ($a$), the per contact transmission probability mosquito-to-bird ($c$), the mortality rate ($\mu_A$) and the rate with which mosquitoes move from latently infected to infectious ($k$), are temperature ($T$) depended. The per contact transmission probability bird-to-mosquito ($b$), the bird recovery rate ($g$), and the bird mortality rate ($\mu_V$) are constants.

$1/\mu_A$ is the temperature-dependent lifespan of mosquitoes (*lf*). We considered all mosquito-parameters (a, c, 1/*k* and $\mu_A$) to be temperature (T) dependent.

Here, bird-to-mosquito transmission probability or vector competence (*c*) is considered a product of infection efficiency (the proportion of mosquitoes that develop infection), and transmission efficiency (the proportion of mosquitoes that become infectious) [30]. The vector-host ratio is considered as the total number of mosquitoes divided by the total number of birds [$(N_M)/(S_B + I_B + R_B)$]. We recorded the cumulative incidence of WNV cases in birds using a dummy compartment.

The infection rates per susceptible bird (*B*) and susceptible mosquito (*M*) are given by Eqs 1 and 2 [37], where $N_B$ and $N_M$ are the total population size of birds and active mosquitoes; $I_B$ and $I_M$ the number of infectious individuals of each group.

$$B = \frac{I_M}{N_B} \cdot a \cdot b \cdot d \tag{1}$$

$$M = \frac{I_B}{N_B} \cdot a \cdot c \cdot d \tag{2}$$

We took population dynamics into account by incorporating a mortality rate and an equal birth rate for both mosquitoes and birds that resulted in a constant population size. The basic reproduction number ($R_0$) of this model for the period when mosquitoes are active, is provided in Eq 3 (see [31] for the derivation).

$$R_0 = \sqrt{\frac{a \cdot b \cdot d}{\mu_A} \frac{a \cdot c \cdot d \cdot \frac{S_M0}{N_B0} \cdot \frac{k}{k+\mu_A}}{\mu_V + g}} \tag{3}$$

## Parametrization and data

We based the parameters of the model on observations from laboratory experiments and literature (Table 1). For the temperature dependent parameters, we used temperature-response curves and their parameters were based on Shocket et al. (2020) [27]. There, the authors fit the temperature-response curves following linear, Briére, or quadratic functions (S2 Text).

Rates in days$^-$1, [T] = temperature dependence. Based on: [27]. Biting rate: Brière, with $q$ = 1.70E-4, $T_{min}$ = 9.4, $T_{max}$ = 39.6 incubation rate: Brière, with $q$ = 7.38E-5, $T_{min}$ = 11.4, $T_{max}$ = 45.2; *c* quadratic with $q$ = -3.05E-3, $T_{min}$ = 16.8, $T_{max}$ = 38.9. Data from: [19–26].

The remaining parameters (dilution, vector-host ratio, onset/duration mosquito season) were set differently according to the model set and outcome observed, and are explained in the

**Table 1. Set of parameters, values and their source used to model West-Nile Virus (WNV) dynamics between mosquitoes and birds.**

| Parameter | Symbol | Value/range/formula | Reference |
|---|---|---|---|
| Biting rate mosquito [T] | *a* | Briére (T, q, tmin, tmax) | [27] |
| Lifespan mosquito [T] | *lf* | 169.8–4.86*T days | [27] |
| Mortality rate mosquito [T] | $\mu_A$ | 1/lf | calculated |
| Virus incubation rate mosquito [T] | *k* | Briére (T, q, tmin, tmax) | [27] |
| B->M transmission probability [T] | *c* | Quadratic (T,q,tmin,tmax) | [27] |
| M->B transmission probability | *b* | 0.8 | [34] |
| Mortality rate birds | $\mu_V$ | 1/550 | [31] |
| Recovery rate birds | *g* | 0.3 | [34] |

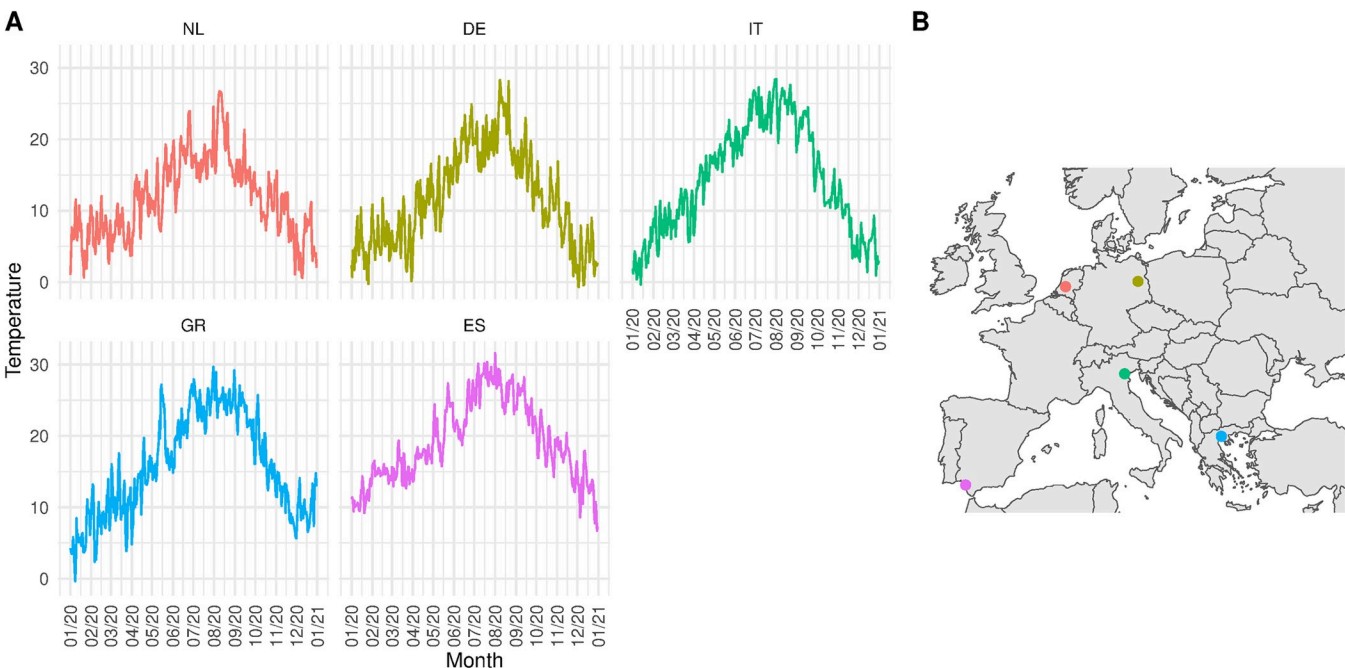

**Fig 2. Daily average temperature for the five locations.** A. Average daily temperature in degrees Celsius for 2020 for the Netherlands [NL] (Utrecht), Germany [DE] (Berlin), Italy [IT] (Po Valley) and Greece [GR] (Aksiou Delta), Spain [ES] (Doñana National Park). B. Geolocated points of the five selected locations. Base layer maps are provided by Natural Earth under CC0 license (https://www.naturalearthdata.com/about/terms-of-use/).

next section (Model implementation and outcomes). To inform the temperature-dependent parameters, we obtained gridded mean daily temperature data based on a 0.1 regular grid from Copernicus [38]. Based on coordinates of the five locations for which we compared the model, we extracted the temperature as time-series from 1980 to 2022. Fig 2 shows the average daily temperatures for the year 2020.

## Model implementation and outcomes

**Implementation 1 (Calculating $R_0$ and the number of days $R_0 > 1$).** We first assessed the $R_0$ (according Eq 3) for the combinations of 16 values of vector-host ratio = $1.5^n$, $n \in \{0: 15\}$, four dilutions (0.25, 0.5, 0.75, 1), and 500 values for the temperature = $0.1 + (n-1) \cdot 0.1$, $n \in \{1:500\}$. We further calculated the number of days in which the $R_0$ is above 1 using vector-host ratio = 100, dilution = 0.5, and daily temperatures (see Fig 3) for all five regions during 1980–2022.

**Implementation 2 (Thresholds for viral circulation).** Next, we assessed which combinations of the dilution, vector-host ratio and onset parameters would lead to a successful viral circulation. We defined a threshold for circulation as the cumulative incidence of infection in birds to be between 5 and 50 cases, with a population size of 1000 birds. Thus, we considered model iterations where the cumulative bird incidence reached these values to have met the boundary conditions for circulation. Here, we assumed that the introductions of infectious birds always took place on the first day of the mosquito season with a proportion of 0.05, or 5 in 1000 birds that become infectious, which were not counted in the cumulative incidence. We propagated the uncertainty on the parameters vector-host ratio, dilution and onset parameters from uniform distributions (Table 2). For each value drawn from the uniform distributions the model was run for one year (2020) in the five locations and the last decade for the

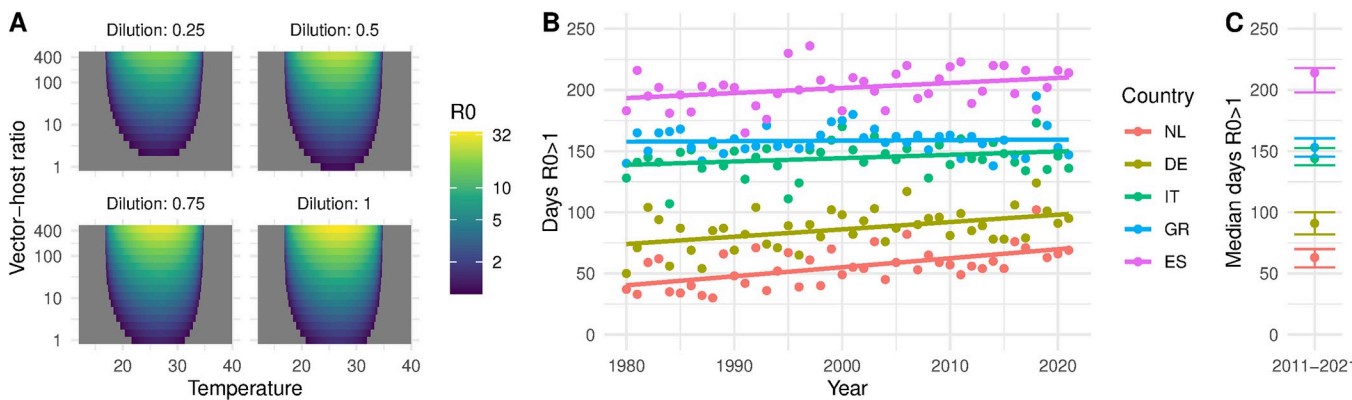

**Fig 3. A.** Relationship between the temperature in degrees Celsius (x-axis), vector-host ratio (y-axis) and dilution (panels) on the $R_0$ (fill-colour). **B.** Median number of days and interquartile range where $R_0>1$ over the period 2011–2021 per location; **C.** Number of days per year with an $R_0>1$ given a vector-host ratio of 1:100, and a dilution of 1:2. Locations: The Netherlands [NL] (Utrecht), Germany [DE] (Berlin), Italy [IT] (Po Valley) and Greece [GR] (Aksiou Delta), Spain [ES] (Doñana National Park).

Netherlands, using average daily temperatures (note, after the parameters were drawn the model behaves deterministically). The model was implemented in *Rstan* (version 2.21.7) running 5000 iterations per location and year. To visualize the threshold as relationship between vector-host ratio and dilution we fitted a linear model to the log converted vector-host ratio and dilution.

**Implementation 3 (Probability of outbreak).**   In the third simulation we used a Gillespie stochastic simulation algorithm allowing for a deterministic ordinary differential equations model to be transformed into a stochastic version of the same model. The Gillespie algorithm determines which of the possible transitions withing the compartmental model takes place and at what time [39]. The 'total rate' is the sum of each transitions event rates and it is used as the parameter for an exponential distribution, determining randomly the time until the next transition event takes place. The probability that an event happens is given by the rate of one event (i.e., infection) divided by the total rate. We ran all the combinations of parameters listed below (Table 2) for the five locations for the year 2020; For Utrecht, we ran the scenarios for the years 2011 to 2022. Again, we assumed that the introductions of infectious birds always took place on the first day of the mosquito season with the introduction of either 2 or 5 infected birds.

We defined an outbreak threshold as at least 5% of the birds to have been infected after the end of the simulation at day 174 (October 1st). Additionally, as a threshold for detection in

**Table 2. Set of parameters, values and their source used to model West-Nile Virus (WNV) dynamics between mosquitoes and birds for simulations 2 and three.**

| Parameter (symbol) | Explanation | Distribution or values | |
|---|---|---|---|
| | | Implementation 2 | Implementation 3 |
| Dilution (d) | Proportion of mosquitoes' bites on WNV-competent birds. | ~U(0.001; 1) | 0.5 |
| Vector-host ratio (VHR) | Number of mosquitoes per bird. | ~U(1; 300) | $2 \cdot (2^{n-1})$, $n \in \{1:7\}$ |
| Season onset | Starting of the mosquitoes season in days after 1st January. | ~U(100; 180) | Set by the time of introduction (t) |
| Season duration | Length of the season in days. | ~U(100; 200) | 174 |
| *Temperature ˚C | Average daily temperature. | Daily variation | Daily variation |
| Number of introductions (I) | Number of infectious birds introduced on t. | 5 | 2 and 5 |
| Time of introduction (t) | Day, from 1st January when the infectious birds were introduced. | First day of season | $110+(n-1) \cdot 10$, $n \in \{1:10\}$ |

~U(): Uniform distribution with parameters minimum and maximum. *Daily variation according to Fig 3.

mosquitoes, we assessed at which time-point mosquito prevalence would surpass a threshold of 1%, which translates to a required sample size of 300 mosquitoes to detect at least 1 infected mosquito with 95% certainty (see S3 Text). Additionally, we calculated the required sample size to detect at least one infected mosquito with a certainty of 95% based on the peak prevalence in mosquitoes. We summarized the iterations per combination of vector-host ratio, time of introduction and year of simulation as the median required sample size. For each scenario, we ran 1000 iterations of the model and summarized the proportion of iterations that resulted in an outbreak. The model was implemented in *SimInf [40]* and we used the '*SimInf events*' function within the package to introduce infections. In all implementations of the model, we assume that the vector-host ratio remains constant throughout the season.

## Results

### $R_0$ and the number of days $R_0 > 1$

We explored the theoretical relationship between temperature and the basic reproduction number ($R_0$) (Fig 3A). When we calculated the daily $R_0$ based on the observed daily average temperatures of the five locations, we found that the average number of days per year in which WNV outbreaks can take place ($R_0 > 1$), given that the vector-host ratio is 100 and the dilution is 0.5, has been overall inclining over the last 40 years (Fig 3B). Of the five locations that we considered, the Netherlands has 30% of the number of days with an $R_0 > 1$ compared to Spain between 2011–2021 (Fig 3C).

### Thresholds for viral circulation

When we assessed under which combination of dilution, vector-host ratio and mosquito season onset and length, given the observed daily temperatures circulation took place, we found that the temperature in Greece, Spain and Italy allowed for circulation under low vector-host ratios, and at high dilutions. For example, with 50% of birds susceptible and competent (dilution = 0.5), circulation was already occurring at vector-host ratios of 1:10 (Fig 4). In the

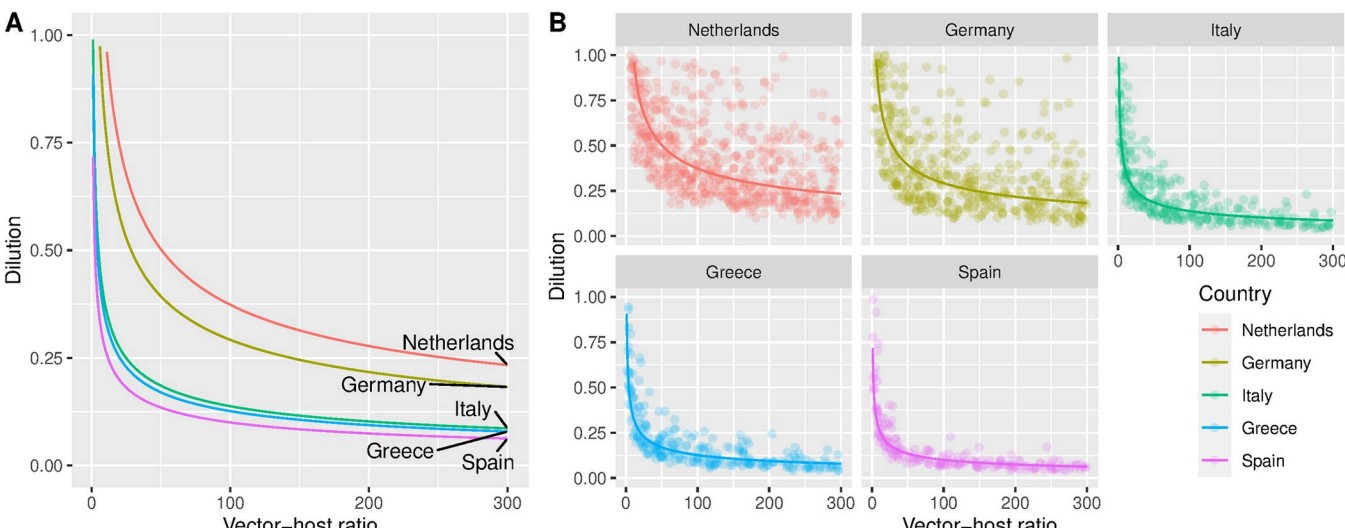

**Fig 4. A.** Average threshold for WNV circulation obtained combining vector-host ratio and dilution for five locations in 2020: The Netherlands (Utrecht), Germany (Berlin), Italy (Po Valley) and Greece (Aksiou Delta), Spain (Doñana National Park). The lines represent the linear fitted relationship between the 10-log converted points; **B.** Thresholds obtained in each model iteration (points) and the fitted average (solid line) that resulted in the transmission of WNV given a vector-host ratio and dilution. The threshold was defined as a cumulative incidence in birds of between 5 and 50 cases at the end of the year.

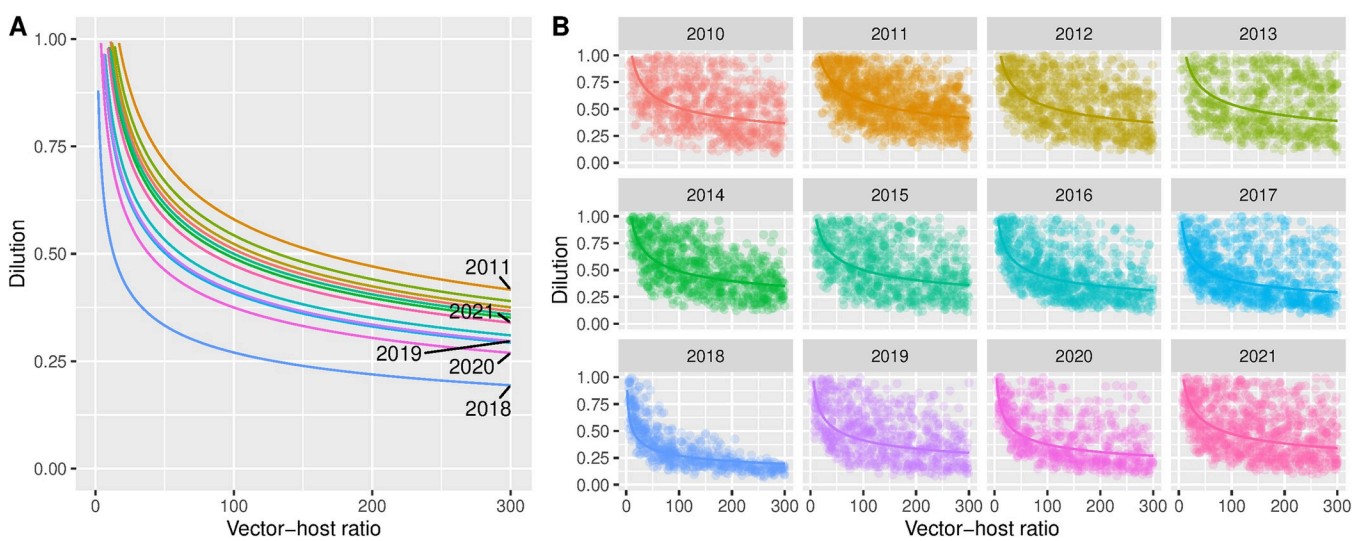

**Fig 5. A.** Average threshold for WNV circulation obtained combining vector-host ratio and dilution between 2011–2021 for Utrecht (the Netherlands). The lines represent the linear fitted relationship between the 10-log converted points; **B.** Thresholds obtained in each model iteration (points) and the fitted average (solid line) that resulted in the transmission of WNV given a vector-host ratio and dilution from 2010–2021 for Utrecht (the Netherlands).

Netherlands and Germany, thresholds for circulation were higher. There, larger proportions susceptible birds (lower dilutions) and higher vector-host ratios were required. In the Netherlands, circulation seems to be seldom at vector-host ratios below 1:100 at a dilution of 0.5.

Similarly, a marked variation between years occurred. For the Netherlands (Utrecht), the threshold for circulation was reached at lower levels of vector-host rate and dilution in 2018 compared to the other years of the same decade (Fig 5).

## Probability of outbreak

We found that the probability that introductions result in a WNV outbreak, in a fully susceptible population, depends on the timing of the introduction. For the Netherlands, a short window of introductions between late May and mid-June would result in outbreaks (Fig 6). For the other locations, the window was wider, and outbreaks occurred at lower vector-host ratios (S1A Fig). In Spain, Italy and Greece, introductions would already lead to a 1% prevalence in mosquitoes starting from July for high vector-host ratios, to August for lower vector-host ratios (S1B Fig). However, due to previous circulation in these regions, the assumption of a fully susceptible population only serves as counterfactual for the Netherlands and does not reflect reality there. In the Netherlands and Germany, only late (mid-August) circulation reached that threshold (Figs 6B and S2B). There, we observed marked differences between years as well. In 2018, the conditions under which circulation takes place, and the length of the transmission-season allowed for more WNV circulation compared to other years. For the Netherlands, in years with limited transmission potential such as 2021, we would require a minimal sample size larger than 1000 to detect at least one infected mosquito given a high vector-host ratio and an optimal introduction date (Fig 6C); for many other combinations of the introduction dates and vector-host ratios we would need sample sizes larger than 3000.

## Discussion

Here, we showed that the temperate maritime climate of the Netherlands only seems to allow for WNV circulation during warmer summers and in settings with a high proportion of

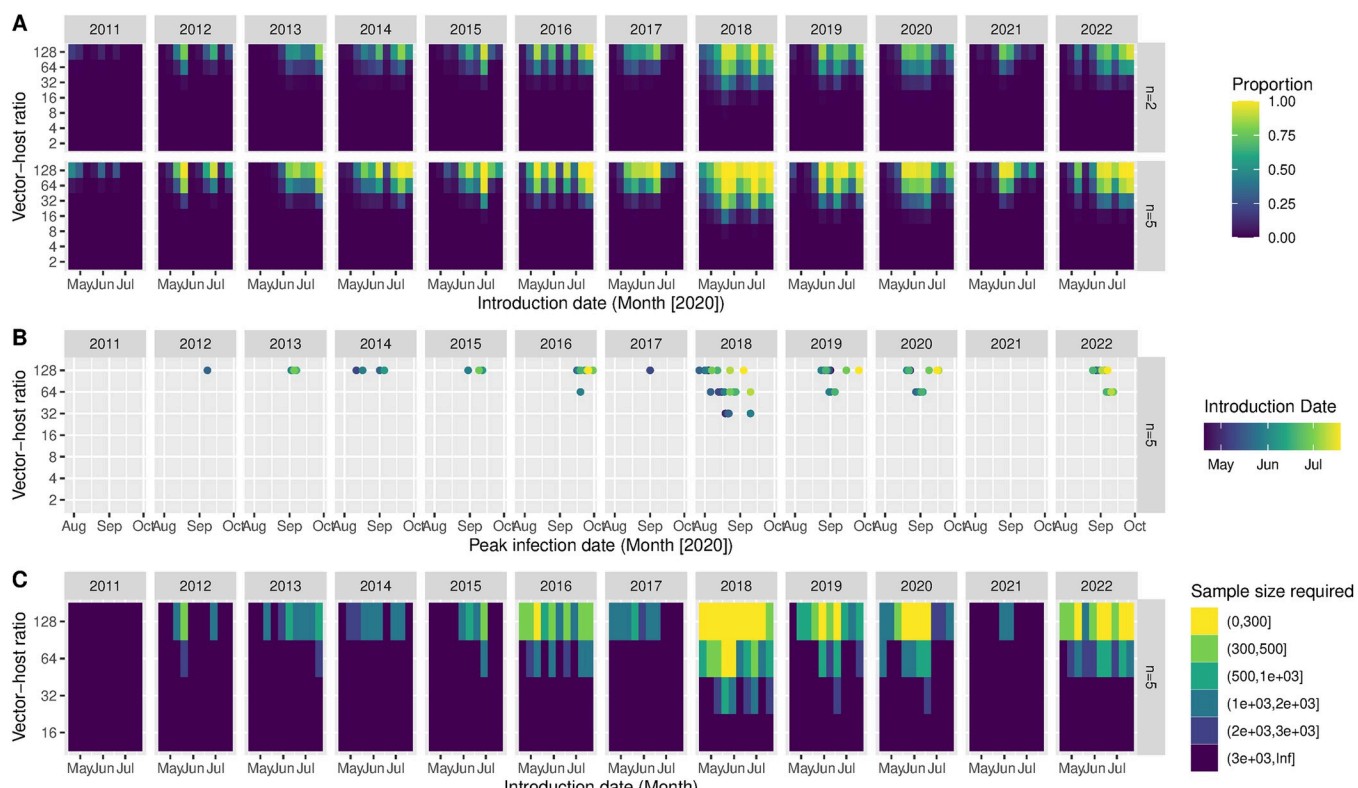

**Fig 6. A.** The proportion of introductions that result in a WNV outbreak by introduction date (x-axis) and vector-host ratio (y-axis); for five locations (facet columns) and with an introduction of 2 or 5 of infected birds (facet rows, n = 2, n = 5) in the Netherlands (Utrecht) in a fully susceptible population. **B.** Date on which the proportion of infected mosquitoes surpasses 1% (x-axis) after introductions at different vector-host ratios (y-axis) and introduction dates (colour) for the Netherlands (Utrecht) by year (facets). **C.** Sample size required (colour) to detect at least one positive mosquito with a certainty of 95% at the peak prevalence in mosquitoes, at different vector-host ratios (y-axis) and introduction dates (x-axis).

competent hosts, and a high vector-host ratio. The temperate continental climate of the area around Berlin forms a more suitable setting for circulation. We observed an increase of the number of days in which circulation of WNV is possible over the last decades. In the Mediterranean region, where WNV has been present for at least several decades, temperature is hardly a limiting factor on WNV circulation. We found between-year variation, where the model results indicated that 2018 was indeed an exceptionally suitable year for WNV circulation, where both the probability of circulation was higher and suitable period for circulation was longer.

In the context of risk-based surveillance in mosquitoes and (wild) birds [30], these results inform the setting and time-span that we need to focus on for regions such as the Netherlands: We expect late summer circulation, only in settings where there is a large proportion of competent birds and high vector-host ratios. This means that primarily areas rich in competent species with high vector-host ratios should be considered for surveillance, and that circulation is more likely to be detected in August and September. This aligns with the detection of WNV in birds and mosquitoes in the Netherlands in 2020 and in the grey heron in 2022 [14,15]. We showed that the detection of WNV in mosquitoes in the Netherlands would often requires large sample sizes, since low circulation occurs. Other, often spill-over host species such as wildlife and livestock might thus serve as better sentinel populations for surveillance. It should be noted however, that the suitability of sentinel species differs per region and is context dependent due to differences in human and animal population density, accessibility,

vaccination status of animals, and past/co-circulation of flaviviruses amongst other factors. For example, sentinel chickens seroconverted prior to cases of WNV in humans in Greece and Italy [41,42], while evidence of WNV circulation was proven in feral horses before any human cases appeared in multiple areas in Spain [4,43].

From a One Health perspective, it is important to be able distinguish between sylvatic WNV circulation, where the virus is maintained between wild birds and ornithophilic mosquitoes and the synanthropic circulation, where mosquitoes feed and transmit both to birds and humans. Here, we did not take into account human or equine spill-over, as this is dependent on multiple factors for which comparable data are lacking at a European scale. Examples of such data types are the composition on local mosquito populations and their biting behaviour, but also the human population density and their behaviour. In a setting with primarily ornithophilic mosquitoes, WNV might circulate but rarely spills-over. In the Netherlands in 2020, six human cases were detected while no equine cases were reported. This might be explained by the host-preference of the local mosquito populations. Thus, understanding local vector composition [44], competence [17], biting behaviour [45] and overwintering potential [46] are essential to produce models at a smaller spatial scale, and model spill-over to humans and horses.

Compared to other temperature-dependent mathematical models of WNV transmission, we considered the uncertainty around the vector-host ratio and abundance of competent birds (dilution). Often modelling approaches either incorporate temperature dependence in calculating transmission potential or suitability as a function of temperature [32,34,35], but seldom incorporate temperature-dependence of parameters. Vogels et al. (2017) explicitly took into account the presence of multiple *Cx. pipiens* s.s. biotypes, by incorporating their biting behaviour in their $R_0$ model [34], where we implicitly incorporate this into the dilution. Bhowmick et al (2020) did account for temperature-dependence, but assumed fixed carrying capacities for birds and mosquitoes, and thus fixed vector-host ratios [12]. However, here we did capture the range of properties within the range of the 'dilution' parameter. Once reliable data about temperature is acquired and allows to forecast trends and patterns, the model presented here can be extended to predict the WNV infection dynamics in the future, enabling risk managers to anticipate mitigations strategies. The model could also be used in combination with modelled temperature prediction based on different emissions scenarios and socio-economic pathways (Shared Socioeconomic Pathways, SSP) as modelled by the Intergovernmental Panel on Climate Change [47]. However, predictions should be interpreted with caution, since temperature is only part of the complex relationship between mosquito, virus and environment.

Our model does have several limitations. First, we assumed that the vector-host ratios were constant per season. In reality this is not the case, and overestimating the vector-host ratio early in the season will inflate the probability of transmission. Thus, the results presented here can be considered as 'worst-case' scenarios, since reality will be a composite of different vector-host ratios over time. Understanding what drives mosquito and bird abundance helps to further inform vector-host ratios. However, at a European scale, data is lacking and mosquito abundance models [48] often only approach abundance as peak-abundance and fail to provide temporal changes. Mosquito abundance is driven by different ecological drivers namely light intensity, $CO_2$, temperature, humidity, drainage, and vegetation, depending on the location and local composition of habitats, their carrying capacity and the mosquito population [18,33,49]. Second, we did not perform formal inference. Time-series data providing unbiased estimates of incidence or prevalence of WNV in birds across the different locations were not available. While WNV prevalence in trapped mosquitoes did provide information on the presence of WNV in mosquitoes over time within the catchment area of the traps and linking this data to prevalence in birds proved to be challenging. Birds move in and out of the catchment

area and their exposure can vary. Since WNV circulation can be spatially very heterogeneous, prevalence in birds reflects a combination of movement and transmissibility. Without proper understanding and description of the spatial movement of birds and the availability of appropriate denominator data (abundance of birds), it becomes difficult to distinguish transmission and movement patterns. Especially in Europe, the incidence of WNV in birds is seldom known as clinical signs in birds are rare, except for some species such as the Northern goshawk (*Accipiter gentilis*) [50]. This is further complicated by the fact that birds are a highly heterogeneous group, where even within a single species there are differences in their levels of viraemia and thus their capacity to propagate infection[50]. Third, we considered transmission to occur between '*Cx. pipiens pipiens*' mosquitoes and 'birds', whereas in reality, *Cx. pipiens* s.s. can be distinguished into two biotypes and their hybrids [17]. Distribution and transmission characteristics differ between these types, influencing transmissibility [17,44,51]. Similarly, different species of birds can have varying levels of viraemia, leading to differences in their host-competence statuses.

From these limitations, that were often a result of the absence of appropriate data, we arrive at several recommendations. 1) Evaluate in a local setting with local mosquito populations how temperature affects WNV transmissibility. 2) Develop appropriate mosquito abundance models, adapted to the local setting. 3) Understand local compositions of *Cx. pipiens* s.s., since the biotypes and hybrids show distinct feeding behaviours and have different roles in amplification and spill-over. 4) Increase the understanding of which birds in Europe play a key role in WNV amplification, their distribution and movement patterns. Future extensions of the model would include simulating multiple years and incorporating diapausing mosquitoes will be of interest, however, understanding under which conditions overwintering takes place is crucial [44]. Also, we need to be able to disentangle the contribution of re-introductions from other regions and the local persistence through overwintering.

Future research should also strive to improve the parameterization of the temperature-dependent parameters. This was now based on data that was collected in laboratory settings, under constant temperature regimes often using mosquito populations from the United States of America and the New York 1999 strain of the virus [19–26]. Local mosquito populations might show different thermal optima and responses, which could influence the boundary conditions for circulation. Mosquitoes are known to adapt to local settings, and the virus might as well be able to adapt. It has been shown that transmissibility of the WNV lineage 2 in a Dutch *Cx. pipiens* colony was higher compared to a North American colony [52]. Having access to temperature-trait research specific to local populations and strains, and under fluctuating temperature regimes will increase the reliability of predictions.

## Conclusion

The disease burden of WNV in Europe has been increasing over the last decades and this trend is expected to continue due to climate change. Understanding what drives transmission, and identifying the boundary conditions under which circulation will take place is essential for preparedness. Although other drivers for the abundance of mosquitoes entering in the model (e.g., sun light intensity, $CO_2$, precipitation, humidity, and vegetation) should still be explored in further model implementation, the temperature-dependent WNV transmission model we presented here, will facilitate in assessing the future risk of transmission in northern European countries, where temperature is currently still a limiting factor, and help in planning surveillance to detect when thresholds are reached, thereby improving preparedness efforts.

## Disclaimer

"The views and opinions of the authors expressed herein do not necessarily state or reflect those of ECDC. The accuracy of the authors' statistical analysis and the findings they report are not the responsibility of ECDC. ECDC is not responsible for conclusions or opinions drawn from the data provided. ECDC is not responsible for the correctness of the data and for data management, data merging and data collation after provision of the data. ECDC shall not be held liable for improper or incorrect use of the data".

## Supporting information

**S1 Text. Human West Nile virus cases in Europe.**
(DOCX)

**S2 Text. Temperature dependent West Nile virus transmission parameters.**
(DOCX)

**S3 Text. Mosquito West Nile virus sampling in selected European regions.**
(DOCX)

**S1 Fig. The proportion of introductions that result in a West Nile virus outbreak in the Netherlands, Germany, Italy, Greece and Spain, assuming a fully susceptible population in 2020.**
(TIF)

**S2 Fig. The proportion of introductions that result in a West Nile virus outbreak in Berlin between 2011–2022.**
(TIF)

## Acknowledgments

We thank Dr. Quirine Ten Bosch for her critical review of the manuscript.

Data from The European Surveillance System–TESSy, provided by Austria, Bulgaria, Cyprus, Czechia, Germany, Greece, Spain, France, Hungary, Italy, North Macedonia, Netherlands, Portugal, Romania, Slovenia, Slovakia, Turkey and released by ECDC.

## Author Contributions

**Conceptualization:** Michel Jacques Counotte.

**Data curation:** Mariana Avelino de Souza Santos.

**Formal analysis:** Eduardo de Freitas Costa, Michel Jacques Counotte.

**Funding acquisition:** Michel Jacques Counotte.

**Investigation:** Kiki Streng, Mariana Avelino de Souza Santos, Michel Jacques Counotte.

**Methodology:** Eduardo de Freitas Costa, Michel Jacques Counotte.

**Project administration:** Michel Jacques Counotte.

**Supervision:** Michel Jacques Counotte.

**Validation:** Eduardo de Freitas Costa, Kiki Streng, Mariana Avelino de Souza Santos.

**Visualization:** Kiki Streng, Michel Jacques Counotte.

**Writing – original draft:** Michel Jacques Counotte.

**Writing – review & editing:** Eduardo de Freitas Costa, Kiki Streng, Mariana Avelino de Souza Santos.

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
