## [Decision Letter · Decision Letter 0]

26 Feb 2024

Dear Dr. Counotte,

Thank you very much for submitting your manuscript "The effect of temperature on the boundary conditions of West Nile virus circulation in Europe" for consideration at PLOS Neglected Tropical Diseases. As with all papers reviewed by the journal, your manuscript was reviewed by members of the editorial board and by several independent reviewers. The reviewers appreciated the attention to an important topic. Based on the reviews, we are likely to accept this manuscript for publication, providing that you modify the manuscript according to the review recommendations.

Sincerely,

Song Liang

Academic Editor

Justin Remais

Section Editor

Reviewer's Responses to Questions

**Key Review Criteria Required for Acceptance?**

**Methods**

-Are the objectives of the study clearly articulated with a clear testable hypothesis stated?

-Is the study design appropriate to address the stated objectives?

-Is the population clearly described and appropriate for the hypothesis being tested?

-Is the sample size sufficient to ensure adequate power to address the hypothesis being tested?

-Were correct statistical analysis used to support conclusions?

-Are there concerns about ethical or regulatory requirements being met?

Reviewer #1: (No Response)

Reviewer #2: How did the authors arrive at the total number of mosquitos populating the SEI model? Other ecological factors besides temperature, e.g., precipitation, are drivers of the total number of mosquitoes that begin the transmission cycle and factor into the vector-host ratio.

**Results**

-Does the analysis presented match the analysis plan?

-Are the results clearly and completely presented?

-Are the figures (Tables, Images) of sufficient quality for clarity?

Reviewer #1: (No Response)

Reviewer #2: (No Response)

**Conclusions**

-Are the conclusions supported by the data presented?

-Are the limitations of analysis clearly described?

-Do the authors discuss how these data can be helpful to advance our understanding of the topic under study?

-Is public health relevance addressed?

Reviewer #1: (No Response)

Reviewer #2: Although the model includes only temperature as a driving force for WNV transmission, it may be important to mention in the conclusion that other ecological drivers should be taken into consideration to model the total number of mosquitoes entering the model.

**Editorial and Data Presentation Modifications?**

Reviewer #1: (No Response)

Reviewer #2: (No Response)

**Summary and General Comments**

Reviewer #1: Costa de Freitas et al. modeled the effect of temperature on the boundary conditions of West Nile virus circulation in Europe. The authors used a compartmental model to explore under which conditions West Nile virus transmission can take place in five locations in Europe. These locations varied in their history of West Nile virus transmission, ranging from endemic regions with decades of reported transmission and newly emerging regions. The main findings are that temperature is an important limiting factor to West Nile virus transmission in the newly emerging regions, whereas it is hardly a limiting factor in endemic regions. This work provides important insights to better inform risk assessments and surveillance strategies. 

The two main limitations of the model are the assumptions that (1) mosquito and bird populations are constant over time, and (2) vector-host ratio is constant over the season. The authors did an exceptional good job of acknowledging these limitations, which stem from lack of mosquito and bird abundance data availability, and they provide specific recommendations for how these current data gaps can be addressed in future work. Despite these limitations, the model provides important insights into retrospective transmission patterns in previous years, and the role of temperature. The only thing that in my opinion is still missing is a section in the discussion on how this model can be used/adapted to predict future transmission (i.e., can this approach be used to predict what level of transmission is expected in the following months, which can then be used to inform control strategies to prevent transmission? Or can this only be used for retrospective data?). 

Minor comments

1. The species name should not be capitalized: e.g. Culex pipiens instead of Culex Pipiens.

2. Line 357: for consistency, I suggest changing “subtypes” to “biotypes”.

Reviewer #2: (No Response)

PLOS authors have the option to publish the peer review history of their article (what does this mean?). If published, this will include your full peer review and any attached files.

Reviewer #1: No

Reviewer #2: No

Figure Files:

Data Requirements:

Reproducibility:

References

---

## [Editor Report · Decision Letter 1]

22 Apr 2024

Dear Dr. Counotte,

We are pleased to inform you that your manuscript 'The effect of temperature on the boundary conditions of West Nile virus circulation in Europe' has been provisionally accepted for publication in PLOS Neglected Tropical Diseases.

Best regards,

Song Liang

Academic Editor

Justin Remais

Section Editor

---

## [Editor Report · Acceptance letter]

30 Apr 2024

Dear Counotte,

We are delighted to inform you that your manuscript, "The effect of temperature on the boundary conditions of West Nile virus circulation in Europe," has been formally accepted for publication in PLOS Neglected Tropical Diseases.

Best regards,

Shaden Kamhawi

co-Editor-in-Chief

Paul Brindley

co-Editor-in-Chief
